# Therapeutic Advances in Bladder Preservation for BCG-Unresponsive Non-Muscle Invasive Bladder Cancer

**DOI:** 10.3390/cancers17040636

**Published:** 2025-02-14

**Authors:** Alyssa Lange, SriGita Madiraju, Firas G. Petros

**Affiliations:** Department of Urology, College of Medicine and Life Sciences, The University of Toledo, Toledo, OH 43614, USA; alyssa.lange2@rockets.utoledo.edu (A.L.); srigita.madiraju@utoledo.edu (S.M.)

**Keywords:** bladder preservation, intravesical chemotherapy, immunotherapy, gene therapy, BCG-unresponsive, non-muscle invasive bladder cancer

## Abstract

BCG is standard intravesical treatment for high-risk NMIBC, however, recurrence remains a challenge. New treatments, including immunotherapy, novel agents and devices, offer promising bladder-sparing options for BCG-unresponsive cases.

## 1. Introduction

Bladder cancer is classified into NMIBC or muscle invasive bladder cancer (MIBC) according to the American Joint Committee on Cancer (AJCC) TNM staging. On diagnosis, 75% of the cases are NMIBC (Tis, Ta, T1). NMIBC is further broken down into risk classifications dependent on tumor size, histologic characteristics, multifocality, recurrence, lymphovascular or prostatic urethral involvement, and response to BCG treatment. The primary treatment for intermediate and high-risk NMIBC is transurethral resection of bladder tumor (TURBT) followed by intravesical therapy. Standard intravesical therapies can include chemotherapy or immunotherapy. Chemotherapy agents include mitomycin C, gemcitabine, epirubicin, and docetaxel, which are given as a single dose post-operatively or as adjuvant therapy. The gold standard adjuvant treatment involves intravesical immunotherapy with BCG, an attenuated mycobacterium strain [1]. Despite the initial response to adequate BCG therapy, however, more than 50% of patients with NMIBC have recurrence or progression in 1–2 years and up to 30–40% of them will eventually become unresponsive to BCG [2,3]. The development of BCG-unresponsive disease could be related to tumor biology with differences in tumor microenvironment molecular changes, impaired immune response, or inadequate BCG therapy related to difficulty for patients to adhere to treatment regimens despite the warranted therapy. In a recent study published on 5803 patients with high-risk NMIBC, while 86.9% had adequate BCG induction, only 41.5% had adequate BCG induction and maintenance [4].

The US Food and Drug Administration (FDA) guidelines define BCG-unresponsive disease as persistent or recurrent carcinoma in situ (CIS) alone or with recurrent Ta/T1 disease within 1 year of completion of adequate BCG therapy, recurrent high-grade (HG) Ta/T1 disease within 6 months of finishing adequate BCG therapy or T1 HG disease at the first evaluation following BCG induction [5]. Adequate BCG treatment is defined as minimum five of six doses of the initial induction course plus at least two of three doses of maintenance or two of six doses of a second induction course [5]. Initial induction is performed 3–4 weeks post-TURBT with weekly instillations for approximately six weeks. While there is no standardized protocol, however, the SWOG regimen is most commonly used in the United State. This protocol includes 6-week induction, with 3 weekly instillations at 3, 6, 12, 18, 24, 30, and 36 months. There is controversy regarding the optimal regimen and variations in practice at this time [6]. Ensuring adequate BCG treatment is essential before considering alternative therapies, as patients who have received inadequate BCG may benefit from further BCG treatment prior to exhausting alternative novel or experimental treatment options. Patients with BCG-unresponsive NMIBC are unlikely to benefit from additional BCG. The standard of care for this cohort is RC. Studies have shown RC to have a cancer-specific survival of >80% and is associated with lower metastasis and lower risk of death from bladder cancer within 24 months [7]. However, there remains a subset of patients who choose not to pursue an invasive surgical technique due to concerns of morbidity, mortality, and quality of life. Others are ineligible due to significant medical comorbidities such as cardiovascular diseases or kidney diseases, age, and frailty. In these patients, their only organ-sparing option may include single agent intravesical chemotherapy induction and maintenance, with limited data supporting the use of radiation for T1 NMIBC [8,9].

Identifying alternative therapies for BCG-unresponsive NMIBC has sparked several studies for novel approaches. These novel options will hopefully address limitations including the lack of care standardization for papillary-only BCG-unresponsive NMIBC who are ineligible for or decline RC. Quality of life measures play an important role in decision making for patients who decline cystectomy, but these are difficult to capture. This review aims to explore recent and upcoming advancements in BCG-unresponsive NMIBC treatment and alternative organ-preserving options including their potential in clinical practice.

## 2. Current Organ-Preserving Options for BCG-Unresponsive NMIBC

### 2.1. Single Agent and Combination Chemotherapy Approaches

First, we review the current organ-preserving options for BCG-unresponsive NMIBC patients. Non-immunotherapy investigational agents or therapies that have been utilized for these patients include sequential intravesical valrubicin, gemcitabine and docetaxel (gem/doce), or radiofrequency-induced thermo-chemotherapy effect (RITE) [3,10].

Previously, for 20 years, valrubicin was the only FDA-approved intravesical treatment for BCG-unresponsive CIS, with a recurrence free survival of 18–21% at 6 months and 8% at 30 months.

On the other hand, gemcitabine is a deoxycytidine nucleoside analog and docetaxel is a microtubule depolymerization inhibitor. A multi-institutional study by Steinberg et al. on sequential intravesical gem/doce included 45 patients with only 4/45 being BCG-naïve and found that 54% and 34% of patients had 12 months and 24 months recurrence free rates, respectively, for NMIBC and 66% had complete response (CR) at first surveillance [11]. Concern was noted regarding oncologic safety of delaying cystectomy in the high-risk cohort due to the possibility of disease progression during intravesical treatment. However, 10 patients that proceeded to cystectomy had no positive margins and a low progression rate [11]. Subsequent follow-up with a larger sample size showed that patients with NMIBC previously treated with BCG and subsequently treated with intravesical gem/doce had recurrence-free survival of 60% (165/276) and 42% (116/276) at 12 and 24 months, respectively. Moreover, 43 patients of 276 (15.6%) later underwent cystectomy of which 11 (4.0%) had progression to muscle invasion [3]. Another study examined 33 patients who received the induction of intravesical sequential gem/doce and 21% (7/33) of the patients also had maintenance therapy [12]. BCG-naive, unresponsive/relapsing, and intolerant patients were all included. BCG-intolerant patients were grouped into the unresponsive/intolerant group. However, in the BCG-unresponsive/relapsing patient cohort consisting of 25 patients, 49% 1-year and 34% 2-year HG recurrence-free survival rates were seen. Eight percent (2/25) of additional patients had low-grade Ta recurrence [12]. Limitations of all these studies acknowledged include a retrospective nature, lack of standardization and direct comparison of treatment modalities given varying use of maintenance therapy, different treatment schedules, and heterogeneous patient population between studies.

RITE provides hyperthermia to the bladder wall potentiating cytotoxic effects of chemotherapy and intensifies drug absorption by the formation of tunneling microtubules. However, RITE did not show a significant difference in patients without carcinoma in situ compared to salvage intravesical therapy [10].

We will now review other novel therapies that have been introduced in recent years to address this cohort of BCG-unresponsive NMIBC.

### 2.2. Systemic Treatment

Herein we discuss systemic treatment options for those NMIBC patients who are BCG-unresponsive. Many patients who progress to MIBC are unable to receive standard, first-line cisplatin-based chemotherapy because of renal disease, medical comorbidities, and frailty. The activation of the PD-1–PD-L1 pathway was associated with resistance to BCG in NMIBC. In addition, increased PD-L1 expression was seen in tumors that relapsed post BCG treatment versus BCG-naive tumors. An association of high PD-L1 expression was observed with recurrence and progression [13]. This was initially approved by the European Medicines Agency (EMA) and the FDA for patients with locally advanced or metastatic UC who are cisplatin ineligible, based on KEYNOTE-052 study efficacy and safety data [14]. First-line pembrolizumab was found to have antitumor activity and an acceptable safety/tolerability profile in cisplatin-ineligible patients with urothelial cancer [14]. However, revision was recently made to limit pembrolizumab use in cisplatin-ineligible patients whose tumors express programmed death ligand 1 (PD-L1; combined positive score (CPS) ≥ 10) and to patients ineligible for platinum-based chemotherapy, regardless of PD-L1 status. This was based off the KEYNOTE-361 study, which is a phase III, randomized, open-label trial analyzing pembrolizumab alone or in combination as first-line therapy for advanced UC. The patient criteria required patients with untreated, locally advanced, unresectable, or metastatic UC. While this study found that pembrolizumab plus chemotherapy versus chemotherapy alone did not significantly improve progression-free survival, it did find that patients with PD-L1 expression CPS < 10 who received pembrolizumab had decreased survival versus those who received chemotherapy alone [15]. Long-term outcomes in patients enrolled in KEYNOTE-052 were subsequently analyzed. The objective response rate was 28.6%, with 8.9% and 19.7% achieving complete and partial response, respectively, at the two years follow-up mark with median overall survival being 11.3 months. In patients with CPS ≥ 10, the objective response rate was 47.3% and the median overall survival was 18.5 months [16]. Keynote-057, a phase II, open-label, single-armed study was performed investigating the use of pembrolizumab in patients with BCG unresponsive carcinoma in situ of the bladder with or without papillary tumors. In this study, complete response was achieved in 41% (39/96) of patients treated with pembrolizumab at 3 months. Of the 39 patients, 18 (46%) had CR for 12 or more months with a median follow-up time of 36.4 months. The median progression-free survival was 39.9 months to death, muscle invasive, or metastatic disease. During this study, grade 3 or 4 treatment-related adverse events (AEs) were reported in 13% (13/101), with no treatment-related deaths. This patient cohort also consisted of aging patients with poor prognostic factors or significant comorbidities, which may contribute to a limitation of this study. This prompted FDA approval for pembrolizumab monotherapy in patients with BCG unresponsive, high-risk, NMIBC with CIS with or without papillary tumors who are ineligible for or elected not to undergo cystectomy [17]. Combined positive score has shown to be a helpful assessment to evaluate PD-L1 expression, with a CPS ≥ 10 as a reliable cutoff for pembrolizumab, as mentioned above. This has also helped identify patients who may benefit from adjuvant nivolumab in MIBC [18]. However, the standardization and determination of ideal CPS cutoff for clinical benefit is continuing to be studied and more information is needed concerning its benefit for NMIBC cohorts. The main limitation to the wide-spread use of pembrolizumab in BCG-unresponsive NMIBC is the high rate of immune-related AEs (grade 2 or higher) leading to the early discontinuation of the treatment.

### 2.3. New FDA-Approved Intravesical Gene and Immunotherapies for BCG-Unresponsive NMIBC

Since pembrolizumab, new FDA intravesical therapies for BCG-unresponsive NMIBC have been approved. Herein we discuss nadofaragene firadenovec-vncg (Adstiladrin) and Nogapendekin Alfa Inbakicept-pmln (Anktiva), as shown in Table 1.

Nadofaragene firadenovec (rAd-IFNα/Syn3), otherwise known as Adstiladrin, is made of rAd-IFNα, a non-replicating recombinant adenovirus vector-based gene therapy that distributes the human interferon alfa-2b gene to urothelial cells, and Syn3, a polyamide surfactant that amplifies urothelial viral transduction [2]. Preclinical studies showed that recombinant interferon alfa gene therapy promoted local interferon alfa-2b production and induced tumor regression [19,20]. This prompted a phase 1b trial utilizing intravesical nadofaragene firadenovec in seven NMIBC BCG-unresponsive patients. It was well tolerated with no dose-limiting toxic effects or clinically significant treatment-related AEs, and a single dose was adequate to attain measurable urine interferon alfa levels [21]. Subsequently, a phase 2 trial showed that 35% of patients did not have HG recurrence at 12 months. In this phase 2 randomized study, 43 patients with HG BCG refractory or relapsed NMIBC were included [22]. A phase 3, multicenter, open-label trial was then conducted with 157 patients with BCG-unresponsive NMIBC and an Eastern Cooperative Oncology Group status of 2 or less [2]. Similarly to the phase 2 study, administration protocol started with the initial dose followed by repeat administration at month 3 (day 90), month 6 (day 180), and month 9 (day 270) if no HG recurrence was seen. Moreover, 68% had carcinoma in situ with or without Ta or T1, and 32% had HG Ta or T1 disease alone. All patients had urothelial cell carcinoma with no other histologic variation. A total of 53.4% in the carcinoma in situ cohort had a CR at 3 months and 24.3% remained HG recurrence-free at 12 months. In the HG Ta or T1 cohort, 43.8% were recurrence-free at month 12. It was well tolerated overall with micturition urgency being the most common grade 3–4 adverse event at only 1%, and there were no treatment-related deaths [2]. Based on these results, FDA-approved nadofaragene firadenovec-vncg (Adstiladrin) was the first gene therapy in 2022 for high-risk, BCG-unresponsive NMIBC with CIS with or without papillary tumors. Of note prior to FDA approval, rAd-IFN/SYN3 was granted Fast Track and Breakthrough Therapy Designation (BTD) status in 2018 and subsequently Adstiladrin received Fast Track, BTD, Accelerated Approval, and Priority Review status. The 5-year follow-up efficacy data from a phase III trial demonstrated that for patients with CIS and HG Ta/T1 bladder cancer cohorts, (6/103) 5.8% and (7/48) 15% were HG recurrence–free at month 57, respectively. The Kaplan–Meier-estimated survival at 57 months was 13% and 33% in the CIS and Ta/T1 cohorts, respectively. The 5-year cystectomy-free survival for overall study population was 49%, specifically 43% in the CIS cohort and 59% in the Ta/T1 cohort. Progression to MIBC was noted in five patients (four with CIS and one with Ta/T1) [23].

It was proposed that BCG failure may be partly due to impaired T-cell and cytotoxic cellular responses and the addition of Nogapendekin Alfa Inbakicept (NAI) could act as a second stimulus to enhance BCG synergistically [19]. NAI is an IL-15 super-agonist with the result of the activation and proliferation of NK cells and effector and memory T cells [19]. The QUILT-3.032 study is an ongoing open-label, single-arm trial conducted across 32 clinical sites, with 3 distinct cohorts [19]. Cohort A includes patients with BCG unresponsive CIS with or without Ta/T1 papillary disease treated with NAI plus BCG. Cohort B includes patients with BCG unresponsive HG Ta/T1 papillary NMIBC treated with the same combination. Cohort C initially evaluated NAI monotherapy for BCG-unresponsive CIS but was discontinued after six months due to inferior efficacy compared to NAI plus BCG. Initial findings from the QUILT-3.032 trial showed that NAI combined with BCG offers significant benefits for high-risk CIS disease, with a CR rate of 71%. The treatment involved one instillation per week for six weeks. In responders, 90% avoided cystectomy, with an 89% probability of remaining cystectomy-free at 24 months and 100% bladder cancer-specific survival at the same time point [19]. Notably, female patients had lower CR rates compared to males, with 5 out of 11 females in Cohort A achieving CR compared to 53 out of 71 males in Cohort B. Additionally, no systemic presence of NAI was detected in any patient during the trial [19]. Nogapendekin alfa inbakicept-pmln or Anktiva is currently an FDA-approved immunotherapy as of 22 April 2024 for BCG-unresponsive NMIBC with CIS with or without papillary tumors [24]. Of the 77 study participants qualified to analyze efficacy, 62% achieved CR, with 58% of these remaining NMIBC-free for more than 12 months and 40% NMIBC-free at 24 months [24]. Maintenance therapy was optional in this study and could involve weekly administration for three weeks at months 4, 7, 10, 13, and 19 [19]. However, patients with CIS who do not achieve a CR after the second induction course are advised to consider cystectomy [19,24,25]. ImmunityBio, Inc. released their data analysis as of November 2024 for their ongoing QUILT 3.032 trial in a press release. They reported 100 patients with BCG-unresponsive NMIBC CIS treated with Anktiva achieved a CR rate of 71% with a durable duration of response ranging up to 54 months [26].

## 3. Promising Therapies in Development for BCG-Unresponsive NMIBC

### 3.1. Cretostimogene Grenadenorepvec

Cretostimogene grenadenorepvec (CG0070) is a replication-competent oncolytic adenovirus targeting bladder cancer cells through their defective retinoblastoma (Rb) signaling pathways and therefore potentially expanding the targeted therapy field. CG0070 is an oncolytic adenovirus with an E2F promotor and expresses GMCSF. CG0070 was granted both FDA Fast Track and BTD, highlighting its potential clinical importance and expedited development process. The phase I/II study that enrolled 35 patients showed a tolerable safety profile with patients having Rb pathway defects responding better clinically than their wildtype counterparts (CR 58% vs. 20%). The most common grade 1–2 AE were bladder toxicities, which limited the tolerance of the maximum dosing [27]. The phase II/III BOND study showed an unclear correlation with defective Rb pathways and ended early after CR was 29% after 15 months. Due to tolerability, the International Bladder Cancer group supported continuation for high-risk BCG-unresponsive patients that refused RC—the Bond II study [28]. Interim results from the Bond II study, a phase II single-arm, multicenter trial (NCT02365818) conducted by Packiam et al. showed 45 patients with high-risk BCG-unresponsive NMIBC who refused RC were treated with intravesical CG0070. The results demonstrated a CR of 47% at six months for all patients and 50% for those with CIS. Currently, a phase III, open-label, single-arm trial (BOND-003) designed to evaluate Cretostimogene grenadenorepvec in patients with BCG-unresponsive NMIBC with CIS with or without HG Ta or HG T1 papillary disease (NCT04452591) is occurring. Recent data analysis of the BOND-003 showed a CR rate of 74.5% as of September 2024 with mainly grade 1–2 bladder-related AEs. From the responders, 63.5% and 56.6% continued to show CR for at least 12 and 24 months, respectively. The RC-free and progression-free proportions were 90% and 97.3% at 12 months [29]. Since then, the trial has completed accrual for the CIS cohort and recruitment is still open for the papillary cohort.

Of particular interest to mention is the results of CORE-001, which has also received FDA-BTD, evaluating the synergetic effect of intravesical cretostimogene and pembrolizumab in BCG-unresponsive NMIBC [30]. The trial revealed CR rates in the intention to treat populations at 12 months, any time, and 24 months of 57% (20/35), 83% (29/35), and 46% (16/35), respectively. There were no synergistic AEs. Further studies are warranted to address the substantial unmet need in this space as well as continuing studies to determine treatment longevity. These findings suggest that CG0070 could become an effective emerging alternative either as a monotherapy or in combination with immunotherapy for patients unwilling or unable to undergo RC, with potential approval anticipated as future data becoming available.

### 3.2. TAR-200

An innovative drug delivery system TAR-200 or “The Pretzel” was developed and recently studied as part of the SunRISe-1 trial [31]. This drug was given Breakthrough Therapy Design Designation in December 2023. TAR-200 is an innovative intravesical drug delivery system designed for the sustained release of therapeutics directly into the bladder. It delivers gemcitabine over an extended period, potentially enhancing drug localization and minimizing systemic side effects. The device’s effectiveness is currently being evaluated in the SunRISe-1 trial, a randomized phase 2b study assessing the safety and efficacy of three treatment regimens: Cohort 1 (TAR-200 + cetrelimab, an anti-PD-1), Cohort 2 (TAR-200 alone), and Cohort 3 (cetrelimab alone) [31]. Preliminary results from Cohort 2, which involved 54 patients (30 efficacy-evaluable), show promising outcomes for TAR-200 as a monotherapy. Of these patients, 54% experienced treatment-related AEs, with the most common being pollakiuria, dysuria, and micturition urgency (≥10%). Grade 3 AEs were observed in 7%, and 4% of patients discontinued treatment due to adverse events. No deaths were reported. On the efficacy side, TAR-200 achieved a CR rate in 77% (23 of the 30 evaluable patients), confirmed by cytology and/or biopsy. Of these 23 patients, 21 remained in CR at both 6 months (11/11) and 12 months (6/6) [31]. The sustained local release feature of TAR-200, particularly when combined with therapies like cetrelimab, holds significant potential in bladder cancer treatment, providing a targeted approach that minimizes systemic toxicity while maintaining therapeutic drug levels in the bladder over several days. While preliminary data are promising, continued recruitment will elucidate further feasibility for this device. This innovative drug delivery system could represent a breakthrough in bladder cancer therapy.

## 4. Unapproved FDA Intravesical Therapies

### 4.1. Oportuzumab Monatox-qqrs

Oportuzumab Monatox-qqrs or Vicineum is a recombinant fusion protein designed to block protein synthesis and mediate tumor cell death by selectively targeting epithelial cell adhesion molecule (EpCAM)-expressing cells, which are commonly found in various solid tumors [32]. The drug has been under investigation for its efficacy across multiple solid tumor types. Initial approval was based on a phase 3 single-arm study where patients received 30 mg of Vicineum in 50 mL saline administered intravesically twice weekly for six weeks, followed by once weekly for six weeks. The maintenance dosing was schedule for once weekly every other week for up to 104 weeks [33]. This study demonstrated a CR rate of 40% (36/89) at three months, a median duration of response of 9.4 months, and among responders, 52% remained disease-free for 12 months post-treatment initiation, with a two-year overall survival rate of 96% [33]. The FDA denied Vicineum after initial Priority Review, indicating the additional need for clinical and statistical data and potential drug improvement. There is a phase 3 follow-up trial, a VISTA study, to further investigate FDA concerns, including long-term efficacy, safety, and trial design improvements [34]. Of interest, a phase I single-arm study of the combination of Durvalumab, a human monoclonal antibody (MAb) that inhibits binding of PD-L1, and Vicineum in subjects with HG NMIBC previously treated with BCG is underway. A total of 42% (5/12) of patients had CR after receiving 1500 mg Durvalumab IV every 4 weeks and Vicineum 30 mg, with 58.3% with recurrence and 25% patients experiencing disease progression (NCT03258593) [35].

### 4.2. Inodiftagene Vixteplasmid

BC-819, a DNA plasmid carrying the gene for diphtheria toxin-A regulated by the H19 promoter sequence, had initially been a promising endeavor. This therapy is designed to target cancer cells by leveraging the overexpression of H19, a long non-coding RNA associated with tumorigenesis. A phase 2b trial assessed the efficacy and toxicity of intravesical BC-819 instillations in preventing tumor recurrence and ablating marker lesions. Initial results showed encouraging efficacy including complete tumor ablation in 33% of patients and no recurrence in 64% of the 47 studied patients after three months [36]. However, the phase 2, open-label, monotherapy, single-arm, multicenter clinical trial of BC-819 was terminated due to lack of efficacy (NCT03719300).

## 5. Patient Selection for Organ-Preserving Therapies

Bladder preservation strategies are most suitable for patients who can safely undergo organ-preserving treatments without a high risk of disease progression. However, there is the risk of recurrence and progression increases over time, particularly in patients treated with additional lines of bladder-sparing therapies (BST) [7]. A study found that 32% of patients undergoing BST who experienced recurrence subsequently underwent RC. These patients demonstrated higher rates of node positivity, extravesical disease, and an increased risk of death from bladder cancer, emphasizing the risks associated with delaying RC in this population [7].

When determining the approach, balancing disease control with quality of life outcomes is essential. Survival outcomes differ significantly depending on the patient cohort, with younger, healthier patients deriving more benefit from early RC. While survival outcomes are improved, RC is noted to include a challenging post-operative recovery period, which were mitigated depending on their medical system’s coordination of care, as well as a substantial adjustment period to their daily routines [37]. This difference in survival rates may partly be attributed to the selection of more robust patients for surgery as they are more likely to tolerate and benefit from the invasive procedure [7].

Bladder preservation carries inherent risks, particularly in high-risk patients. These risks include the potential for disease progression to MIBC or metastasis, which can significantly worsen prognosis. Considering these challenges, multidisciplinary care involving urologists, oncologists, radiologists, and other specialists is critical. In our practice, we pay special consideration should a patient experience continued recurrence with equal or more advanced tumor characteristics while attempting to evaluate their individual preference. One good example is progression from BCG-unresponsive NIMBC due to focal CIS with or without Ta component to either diffuse CIS or HG T1 or the involvement of prostatic urethra. Patients’ access to care is another factor influencing our decision in the treatment of BCG-unresponsive NMIBC. Since Adstiladrin is given once every 3 months, it is more convenient for patients traveling for long distances to cancer infusion centers compared to other intravesical therapies required to be given on weekly bases such as Anktiva or gem/doce. When counseling patients with BCG-unresponsive NMIBC, poor tolerance to prior intravesical therapy, bladder storage dysfunction, immune-related AEs of pembrolizumab, and short duration of complete response to a particular treatment are all taken into account, considering radical cystectomy in these circumstances as the primary salvage treatment. Those patients who decline or are ineligible for RC are evaluated for enrollment in clinical trials opened at our center for BCG-unresponsive NMIBC including a (BOND-003) trial for papillary-cohort (NCT04452591), and (TARA-002) trial—Cohort B (NCT05951179), as shown in Table 2. Though ongoing research is needed, authors believe intravesical cretostimogene could become a potentially effective treatment option for BCG-unresponsive NMIBC.

It is essential to evaluate patient comorbidities as these may influence treatment options. For example, patients on immunomodulators for rheumatoid arthritis or inflammatory bowel disease receiving medications such as methotrexate, azathioprine, cyclosporine, sulfasalazine, or receiving biologic agents like TNF inhibitors (adalimumab, etanercept, infliximab), interleukin-6 inhibitors (tocilizumab), and T-cell inhibitors (abatacept) may not mount adequate immune response to intravesical gene and immunotherapies. Although these patients were not excluded from all clinical trials of BCG-unresponsive NMIBC, there is lack of data evaluating subgroup analysis comparing their outcomes to patients who were not receiving such disease-modifying therapies. As such, RC remains their best option to achieve a cure. While some patients may opt for organ-preserving treatments over RC, it is important to recognize that worsening performance status and underlying comorbidities over time could eliminate RC as a viable treatment option. Individualized treatment plans tailored to each patient’s clinical presentation, tumor characteristics, patient’s preferences, insurance coverage, and cost of subsequent treatment help optimize outcomes, ensuring a balance between disease control and the preservation of quality of life [7].

## 6. Rescue BCG as a Concept: Is More BCG Beneficial?

The concept of “Rescue BCG” or additional BCG after initially being unresponsive has emerged as an option for selected patients with BCG-unresponsive NMIBC. With the addition of novel approaches and newly approved agents for BCG-unresponsive NMIBC, is there any benefit to re-introduce BCG? A study conducted at MD Anderson Cancer Center explored the use of additional BCG in this challenging population. Historically, using BCG in this population has become obsolete. Although the findings of their study suggested that some patients could benefit from rescue BCG, the study had a small sample size with limited statistical power, and conclusions about its efficacy were therefore cautious. The researchers emphasized that careful patient selection is essential to identify those who might derive benefits from additional BCG treatments [40].

The study reported that 75% of BCG-unresponsive patients (27/36) achieved disease-free status after receiving rescue BCG. This cohort included patients with HG CIS, T1 disease, Ta disease, and those with CIS alone. The median duration was 83 months. Among patients with CIS with or without papillary disease, 82% (14/17) demonstrated a response at 3 months, although the duration of this response was not reached at a follow-up of 4.7 years [40]. However, rescue BCG is not suitable for all patients. Those with worsening disease, lymphovascular invasion, hydronephrosis, or variant histology were excluded from the study due to the higher risks associated with these features. This highlights the need for rigorous patient evaluation and the development of individualized treatment approaches when considering rescue BCG [40]. The exact mechanism behind the potential beneficial effects of additional BCG in this patient population remains unclear. Given the variability of responses in this study, further research into the specific disease characteristic may offer valuable insights.

## 7. Summary and Future Directions

### 7.1. Summary of Key Findings

This review has provided an overview of both approved and experimental therapies for BCG-unresponsive NMIBC. BCG remains the gold standard for treating high-risk NMIBC, but, unfortunately, many patients develop resistance to this treatment, resulting in recurrence and progression, as well as that documented BCG shortages necessitated rationing BCG usage according to disease risk stratification. As a result, alternative therapies such as intravesical chemotherapy (e.g., gemcitabine, docetaxel) and immune-based therapies (e.g., pembrolizumab, Nogapendekin Alfa-Inbakicep) and vector-based gene therapy (nadofaragene firadenovec) were developed to address BCG unresponsiveness. Among these, pembrolizumab, nadofaragene firadenovec, and Nogapendekin Alfa-Inbakicep have received FDA approval for BCG-unresponsive NMIBC. Despite these advances, significant gaps remain in treatment, particularly concerning the long-term efficacy and safety of these therapies, as well as their cost-effectiveness. Furthermore, there is no current comparison or direction among sequencing these therapies for unresponsive BCG NMIBC. A multidisciplinary approach while taking into some account patient preference is warranted when making these decisions. As BCG-unresponsive NMIBC is a heterogenous disease, depending on disease progression and severity, conversations regarding limitations of BST vs. cystectomy should be made. Clinical trials continue to play a pivotal role in exploring these gaps and refining therapeutic strategies.

Current gaps in treatment include the limited availability of standardized care protocols particularly for patients with papillary-only BCG-unresponsive NMIBC. These patients are either not eligible for RC or opt not to undergo the procedure due to concerns about morbidity and long-term quality of life. Additionally, many of the new and experimental therapies come with varying levels of cost and effectiveness across different patient subgroups, as well as uncertain long-term benefits. The role of clinical trials in advancing care cannot be overstated as they provide essential data on new drugs and treatment combinations, helping to optimize outcomes for diverse patient populations.

### 7.2. Future Research Needs

The future of NMIBC treatment hinges on the completion of rigorous RCTs for new agents and combination therapies. While current studies show promise, large-scale trials are necessary to establish the definitive role of each therapy in clinical practice. Table 2 displays some of the completed and ongoing trials in BCG-unresponsive NMIBC. Emphasis should be placed on exploring the optimal sequencing of treatments, as well as how these treatments can be tailored. These personalization’s would need to address individual patient profiles, including tumor characteristics, molecular subtype of bladder cancer, comorbidities, and treatment tolerance. To our knowledge, no current guidelines or studies define the most appropriate progression of treatment options or specify which treatments should be initiated based on factors such as patient comorbidities, tumor characteristics, and individual patient preferences.

Moreover, there is a strong need for studies that assess long-term outcomes, quality of life, and cost-effectiveness. Many of the current therapies offer short-term relief, but their ability to prevent long-term recurrence and progression, while preserving bladder function, remains uncertain. For example, TAR-200, which delivers sustained drug release directly to the bladder, needs further investigation to confirm their efficacy over extended periods. A larger, well-designed clinical trial utilizing “rescue BCG” could provide a more cost-effective choice in patients that were deemed BCG unresponsive utilizing a standardized definition of BCG-unresponsive NMIBC. As such, long-term follow-up studies are crucial to understanding whether these therapies offer a lasting impact on bladder preservation without compromising patient survival. Furthermore, cost-effectiveness analyses are necessary to ensure that these newer therapies are accessible and feasible in clinical practice, especially given the high costs associated with many of these advanced treatments. Due to the evolving nature of this disease, continued research is warranted.

## 8. Conclusions

The treatment of BCG-unresponsive NMIBC represents a significant challenge, but recent advancements provide hope for more effective organ-preserving options. The potential of combining immunotherapy, gene therapy, and novel drug delivery systems offers an exciting glimpse into the future of bladder cancer treatment. Multidisciplinary treatment strategies will be essential for optimizing outcomes, allowing clinicians to tailor therapy based on individual tumor biology, patient preferences, and overall health. Ultimately, the goal is to provide a broader array of treatment options that will improve survival outcomes while minimizing the impact on patients’ quality of life. As clinical trials progress and new FDA approvals are granted, there is growing potential for longer-lasting benefits for patients with BCG unresponsive disease who are seeking bladder-sparing treatment options.

## Figures and Tables

**Table 1 cancers-17-00636-t001:** Current FDA-approved bladder-preserving treatment for BCG-unresponsive NMIBC.

Agent Type	Route	FDA Approved	Instillation Protocol	Response Rate
Valrubicin	Intravesical	Yes	Induction: 800 mg once a week × 6 weeks Maintenance: not performed	CR 21% at 6 months
Gemcitabine and Docetaxel	Intravesical	No	Induction: Gem 1 g/Doce 37.5 mg once a week × 6 weeks Maintenance: Gem 1 g/Doce 37.5 mg once a month × 12	Recurrence-free survival 60% and 42% at 12 and 24 months, respectively
Pembrolizumab	Intravenous	Yes	200 mg once every 3 weeks × 24 months	CR 41% at 3 months
Nadofaragene Firadenovec-vncg (Adstiladrin)	Intravesical	Yes	75 mL (3 × 10^11^) viral particles per mL every 3 months × 12 months	CR in 53.4% in CIS at 3 months and 24.3% remained HG recurrence-free at 12 months
Nogapendekin Alfa Inbakicept-pmln (Anktiva)	Intravesical	Yes	Anktiva (400 µg) plus BCG (50 mg) per instillation follows BCG SWOG protocol	CR 71% at 54 months

**Table 2 cancers-17-00636-t002:** Current clinical trials for BCG-unresponsive NMIBC.

Study Title	Trial Number	Trial Phase	Novel Therapy	Agent Type	MOA	Complete Response (at Anytime)	Cost/ Year	Status
LEGEND Study: EG-70 in NMIBC Patients BCG-Unresponsive and High-Risk NMIBC Incompletely Treated With BCG or BCG-Naïve	NCT04752722	Phase 1/2	EG-70	Non-viral gene therapy	RIG-1 Agonist + IL-12	71% (15/21)	?	Recruiting
Safety and Efficacy Study of Intravesical Instillation of TARA-002 in Adults with High-grade Non-muscle Invasive Bladder Cancer (ADVANCED-2)	NCT05951179	Phase 2	TARA-002	Lyphilized Group A, type 3 Streptococcus pyogenes Su strain treated with benzylpenicillin	Cell therapy directly kills tumor cells and promotes proinflammatory response by innate and adaptive host immunity	?	?	Recruiting
A Study of Erdafitinib Versus Investigator Choice of Intravesical Chemotherapy in Participants Who Received Bacillus Calmette-Guérin (BCG) and Recurred With High Risk Non-Muscle-Invasive Bladder Cancer (NMIBC)	NCT04172675	Phase 2	Erdafitinib	Pan-fibroblast growth factor receptor FGFR) 1–4 inhibitor	Kinase Inhibitor	?	?	Active, not recruiting
Intravesical Photodynamic Therapy (“PDT”) in BCG-Unresponsive/Intolerant Non-Muscle Invasive Bladder Cancer (“NMIBC”) Patients	NCT03945162	Phase 2	Ruvidar^®^ (TLD-1433) + Photodynamic Therapy (PDT)	Light-activated, ruthenium-based photodynamic compound [38]	Light activation generates cytotoxic singlet oxygen and radical oxygen species	?	?	Recruiting
Atezolizumab in Treating Patients With Recurrent BCG-Unresponsive Non-muscle Invasive Bladder Cancer	NCT02844816	Phase 2	Atezolizumab	anti-PD-L1 inhibitor (monoclonal antibody)	Checkpoint Inhibitor	27 (20/74) [39]	?	Active, not recruiting
Testing the Addition of an Anti-cancer Drug, Pembrolizumab, to the Usual Intravesical Chemotherapy Treatment (Gemcitabine) for the Treatment of BCG-Unresponsive Non-muscle Invasive Bladder Cancer	NCT04164082	Phase 2	Pembrolizumab + gemcitabine hydrochloride	anti-PD1 and antimetabolite deoxycytidine	Checkpoint Inhibitor + antimetabolite	?	$196,588 * + $420 *	Recruiting
IDO and PD-L1 Peptide Based Immune-Modulatory Therapeutic (IO102-IO103) in Combination With Pembrolizumab for BCG-Unresponsive or Intolerant, Non-Muscle Invasive Bladder Cancer	NCT05843448	Phase I	PD-L1/IDO (IO102-IO103) + Pembrolizumab	IDO/PD-L1 peptide based vaccine and anti-PD1	IDO/PD-L1-specific T cell are activated by the vaccine causing a pro-inflammatory space + Checkpoint Inhibitor	?	?+ $196,588 *	Recruiting
A Trial to Evaluate Intravesical Nadofaragene Firadenovec Alone or in Combination With Chemotherapy or Immunotherapy in Participants With High-grade BCG Unresponsive Non-muscle Invasive Bladder Cancer (ABLE-22)	NCT06545955	Phase 2	Nadofaragene firadenovec (Adstiladrin); Nadofaragene firadenovec + gemcitabine & docetaxel; Nadofaragene firadenovec + pembrolizumab	Adenovirus vector based gene therapy carrying rAd-IFNα; + antimetabolite deoxycytidine & microtobule depolymerization inhibitor; + anti-PD1	Gene Therapy Secreting IFN	?	?	Recruiting
Study of Cretostimogene Given in Patients With Non-Muscle Invasive Bladder Cancer, Unresponsive to Bacillus-Calmette-Guerin (BOND-003)	NCT04452591	Phase 3	Cretostimogene (CG0070)	Engineered Oncolytic Adenovirus	Oncolytic Immunotherapy replicates within cells with an impaired retinoblastoma pathway	75% (79/105) [29]	?	Recruiting
Study of CG0070 Given in Combination With Pembrolizumab, in Non-Muscle Invasive Bladder Cancer, Unresponsive to Bacillus Calmette-Guerin (CORE-001)	NCT04387461	Phase 2	Cretostimogene (CG0070) + Pembrolizumab	Engineered Oncolytic Adenovirus + anti-PD1	Oncolytic Immunotherapy replicates within cells with an impaired retinoblastoma pathway + Checkpoint Inhibitor	83% (29–35)	? + $196,588 *	Completed
A Study of TAR-200 in Combination With Cetrelimab, TAR-200 Alone, or Cetrelimab Alone in Participants With Non-Muscle Invasive Bladder Cancer (NMIBC) Unresponsive to Intravesical Bacillus Calmette-Guérin Who Are Ineligible for or Elected Not to Undergo Radical Cystectomy (SunRISe-1)	NCT04640623	Phase 2	TAR-200 + Cetrelimab; TAR-200 Alone; Cetrelimab Alone	Gemcitabine-Releasing Intravesical System +/− anti-PD1	Gemcitabine Delivery System +/− Checkpoint Inhibitor	84% (71/85)	?	Active, not recruiting
QUILT-3.032: A Multicenter Clinical Trial of Intravesical Bacillus Calmette-Guerin (BCG) in Combination With ALT-803 (N-803) in Patients With BCG Unresponsive High Grade Non-Muscle Invasive Bladder Cancer	NCT03022825	Approved	Nogapendekin alfa inbakicept-pmln (N-803) + BCG	IL-15 receptor agonist	IL-15 + BCG	62% (48/77)	644,400 *	Active, not recruiting
ADSTILADRIN (=INSTILADRIN) in Patients With High-Grade, Bacillus Calmette-Guerin (BCG) Unresponsive Non-Muscle Invasive Bladder Cancer (NMIBC)	NCT02773849	Approved	Nadofaragene firadenovec (Adstiladrin)	Adenovirus vector based gene therapy carrying rAd-IFNα	Gene Therapy Secreting IFN	51% (50/98)	$240,000 *	Completed
Study of Pembrolizumab (MK-3475) and Pembrolizumab With Other Investigational Agents in Participants With High Risk Non-muscle Invasive Bladder Cancer (MK-3475-057/KEYNOTE-057)	NCT02625961	Phase 2	Pembrolizumab; Pembrolizumab/vibostolimab coformulation; Favezelimab/pembrolizumab coformulation	anti-PD1; anti-PD1 and anti-TIGIT; anti-LAG-3 and anti-PD1	Checkpoint Inhibitor	41% (39/96)	$195,588 * +/− ?	Recruiting

?: cost is unknown currently; *: estimated cost/year.

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
