# Peer review of "Therapeutic Advances in Bladder Preservation for BCG-Unresponsive Non-Muscle Invasive Bladder Cancer"

_cancers, 2025, doi:10.3390/cancers17040636_

Round 1
Reviewer 1 Report
Comments and Suggestions for Authors
This review article provides a comprehensive overview of current and emerging therapies for BCG-unresponsive non-muscle invasive bladder cancer (NMIBC). The manuscript is well-structured, covering a wide range of treatment modalities, including chemotherapy, immunotherapy, gene therapy, and novel drug delivery systems. The authors have done a commendable job summarizing recent clinical trial results and highlighting the potential of these therapies in clinical practice.
I believe that the study has sufficient merit to be considered for publication on Cancers, although major revisions are required.
MAJOR COMMENTS
- Critical appraisal of evidence: While the review covers an impressive range of therapies, it could benefit from a more critical appraisal of the underlying evidence. A more detailed analysis of key clinical trials, including study design, sample size, and follow-up duration, would strengthen the review's scientific rigor. For instance, when discussing therapies like nadofaragene firadenovec or pembrolizumab, a more in-depth examination of the pivotal trials that led to their approval would provide valuable context for readers.
- Clarity in differentiating therapeutic options: A clearer distinction between approved therapies with established efficacy and those still under investigation would enhance the review's clinical utility. This would help readers better gauge the level of evidence supporting each treatment and make more informed decisions in their practice. Perhaps the authors could incorporate a classification system or visual cues to highlight the approval status of each therapy. I recommend these references that I think are important and that can be of great help when modifying the manuscript (doi: 10.23736/S2724-6051.24.05876-2; doi: 10.1016/j.urolonc.2024.11.015)
- Expansion of patient selection considerations: The section on patient selection, while relevant, could be enriched with a more detailed exploration of the factors influencing treatment decisions. This includes a nuanced discussion of tumor characteristics, patient comorbidities, performance status, and individual preferences. A more comprehensive approach to patient selection would better reflect the complexities of clinical decision-making in this patient population.
- In-depth analysis of "rescue BCG": The concept of "rescue BCG" warrants a more thorough examination. The authors could delve deeper into the rationale behind this approach, exploring the potential mechanisms of delayed response to BCG and the factors that might predict its success. A more balanced discussion of the potential benefits and risks of "rescue BCG" would provide a more complete picture for readers.
- Addressing limitations and future directions: A more explicit discussion of the limitations of the available evidence and the need for further research would strengthen the review's scientific integrity. Highlighting areas where larger, well-designed clinical trials are needed would guide future research efforts and underscore the evolving nature of this field.
The English could be improved to more clearly express the research
Author Response
Please view attached authors response to reviewers comments.
Best regards,
Firas Petros

Reviewer 2 Report
Comments and Suggestions for Authors
This review comprehensively includes the novel and developing therapies for patients with BCG-unresponsive NMIBC. I would like to ask the authors to modify some points as follows.
1. Line 30
Please correct NIMBC to NMIBC.
2. Line 38
The authors stated that most patients with NMIBC become unresponsive to BCG, but this is too much. The authors should note the proportion of BCG-unresponsive in patients who receive adequate BCG therapy.
3. 2.2 Systemic treatment
The authors mentioned the results of KEYNOTE-052 and KEYNOTE-361. The authors should delete this part since these trials were for patients not with BCG-unresponsive NMIBC but with locally advanced / unresectable or metastatic UC.
4. Chapters 7–9 should be changed to 8–10 including subchapters.
Author Response

(The authors gave the same response as above.)
